# *Leptospira* transcriptome sequencing using long-read technology reveals unannotated transcripts and potential polyadenylation of RNA molecules

Ruijie Xu,[1,2] Dhani Prakoso,[3] Liliana C. M. Salvador,[1,2,4] Sreekumari Rajeev[3]

**ABSTRACT** Leptospirosis, caused by the spirochete bacteria *Leptospira*, is an emerging zoonosis that causes life-threatening disease in humans and animals. However, a comprehensive understanding of leptospiral RNA profiles is limited. In this study, we sequenced and analyzed the transcriptome of multiple *Leptospira* strains using Oxford Nanopore Technologies' direct cDNA and direct RNA sequencing methods. We identified new operons, RNA molecules, and evidence of potential posttranscriptional polyadenylation in *Leptospira* transcriptomes. Some RNA molecules that had not been previously annotated could be potential sRNA or noncoding RNA molecules that play a role in gene expression regulation in *Leptospira*. Interestingly, the majority of the new RNA molecules identified in this study were not detected in the nonpathogenic *Leptospira*, suggesting potential virulence-related functions for these molecules. Overall, our study highlights the utility of Oxford Nanopore Technologies's sequencing in studying prokaryotic transcriptome profiles and offers a tool to improve our understanding of prokaryotic RNA landscapes and polyadenylation. Nonetheless, the findings from our study also warrant that the presence of homopolymers of adenine bases in the transcripts may interfere with the interpretation of bacterial transcriptome profiles. Carefully designed experiments are needed to explore the role of the features described in this study in *Leptospira* virulence and pathogenesis.

**IMPORTANCE** Leptospirosis, caused by the spirochete bacteria *Leptospira*, is a zoonotic disease of humans and animals, accounting for over 1 million annual human cases and over 60,000 deaths. We have characterized operon transcriptional units, identified novel RNA coding regions, and reported evidence of potential posttranscriptional polyadenylation in the *Leptospira* transcriptomes for the first time using Oxford Nanopore Technology RNA sequencing protocols. The newly identified RNA coding regions and operon transcriptional units were detected only in the pathogenic *Leptospira* transcriptomes, suggesting their significance in virulence-related functions. This article integrates bioinformatics, infectious diseases, microbiology, molecular biology, veterinary sciences, and public health. Given the current knowledge gap in the regulation of leptospiral pathogenicity, our findings offer valuable insights to researchers studying leptospiral pathogenicity and provide both a basis and a tool for researchers focusing on prokaryotic molecular studies for the understanding of RNA compositions and prokaryotic polyadenylation for their organisms of interest.

**KEYWORDS** transcriptome, long-read sequencing, operons, polyadenylation, prokaryotes, *Leptospira*

Leptospirosis, caused by the spirochete bacteria *Leptospira*, is recognized as one of the most widespread zoonotic diseases that can be equally fatal to both humans

Address correspondence to Sreekumari Rajeev, srajeev@utk.edu, Liliana C. M. Salvador, lilianasalvador@arizona.edu, or Ruijie Xu, Ruijie.xu25@uga.edu.

Liliana C. M. Salvador and Sreekumari Rajeev contributed equally to this article.

The authors declare no conflict of interest.

and animals. Leptospirosis is a neglected disease, estimated to cause over 1 million annual human clinical cases and over 60,000 deaths (1). Pathogenic *Leptospira* spp. are maintained in the renal tubules of asymptomatic animal reservoirs and in contaminated environments such as soil and water (2–4). Recent molecular classification has subdivided the *Leptospira* genus into four subclades (the pathogenic P1 and P2 subclades and the nonpathogenic S1 and S2 subclades) using whole genome sequence data (5). Previous studies suggest that the mechanisms responsible for leptospiral pathogenesis and environmental/host adaptation involve both the loss/acquisition of functional genes at the DNA level and changes in the overall gene expression profiles at the RNA level (6–10). The mechanisms of bacterial gene expression regulation, which govern a wide range of physiological processes, are now understood to be more intricate than previously believed (11). Yet, our current understanding of leptospiral RNA transcription and its regulation mechanisms remains minimal.

The advent of multiple RNA sequencing technologies has paved the way for studying gene expression patterns through the evaluation of bacterial RNA profiles (12). Two long-read sequencing methods, namely, PacBio's single-molecule real-time sequencing and Oxford Nanopore Technologies (ONT) sequencing, have emerged as potent RNA sequencing tools that address some limitations of short-read sequencing (13, 14). Although both technologies generate full-length single-molecule RNA transcripts, ONT sequencing offers the capability to directly sequence cDNA and RNA molecules without the need for fragmentation and PCR amplification, therefore alleviating sequencing biases associated with read-length and primer selection (15–17). Furthermore, the direct RNA sequencing (DRS) method from ONT can sequence native RNA molecules without reverse transcribing RNA molecules into cDNA, preserving posttranscriptional RNA modification markers (14, 18) crucial for gene expression regulation mechanisms (19–21).

Prokaryotic RNA profiles consist of ribosomal RNAs (rRNAs), the major component of the protein-synthesizing apparatus, and transfer RNAs (tRNAs), which transfer amino acids to ribosomes during translation (22–24). Together, they make up over 95% of the entire prokaryotic RNA profile. The messenger RNA (mRNA) transcripts, which contain the open reading frames coding for functional proteins, account for less than 5% of the transcripts (22, 23). In addition, a large variety of noncoding RNA molecules that lack a protein coding sequence are pivotal in modulating prokaryotic gene expression by inhibiting, activating, or suppressing the transcription, translation, or degradation of RNA transcripts (25–27).

Polyadenylation of adenine bases at the 3′ end of RNA transcripts, known as the poly(A) tail, is a widely recognized posttranscriptional modification feature for transcript stability in eukaryotes. However, the presence and function of this feature in prokaryotes remain understudied (21, 28–31). Hence, sequencing prokaryotic RNAs using both ONT cDNA and DRS sequencing protocols requires a synthetic polyadenylation step before adding the pore-targeting molecules to RNA transcripts during library preparation (32–35). While experimenting with ONT's RNA sequencing methods to evaluate the *Leptospira* transcriptome, we accidentally omitted the polyadenylation step in our initial sequencing procedures, yet unexpectedly obtained abundant transcript reads. Because the pore-targeting molecules could have attached to the native prokaryotic poly(A) tail during library preparation to initiate the sequencing process, we hypothesized that the polyadenylated RNA molecules detected in leptospiral transcriptomes using ONT's long-read sequencing methods were posttranscriptionally modified during gene expression/regulation. In this study, we conducted RNA-Seq experiments on two pathogenic and one nonpathogenic *Leptospira* strains with and without the synthetic polyadenylation step using ONT's cDNA sequencing protocol. Concurrently, we performed DRS without synthetic polyadenylation on a selected set of samples. Here, we report the overall *Leptospira* RNA composition and characteristics utilizing ONT's long-read sequencing method.

## MATERIALS AND METHODS

### *Leptospira* culture and RNA extraction

We used two pathogenic strains [*Leptospira interrogans* serovars Copenhageni (LIC) and lcterohaemorrhagiae (LII)] and one nonpathogenic strain [*Leptospira biflexa* serovar Patoc (LBP)]. The stock cultures used are continuously maintained in Ellinghausen-McCullough-Johnson-Harris (EMJH) liquid medium, supplemented with Difco *Leptospira* Enrichment EMJH (Becton Dickinson, Sparks, MD, USA) at 29°C. A 4-d culture of each strain was used for RNA extractions using a commercial kit (miRNeasy Mini Kit, Qiagen, Hilden, Germany) following the protocol. After quantity and quality assessment using Qubit (Thermo Fisher, Waltham, MA, USA) and NanoDrop (Thermo Fisher), the RNAs were further cleaned and concentrated using RNA Clean and Concentrator (Zymo Research, Irvine, CA, USA). The final assessment of RNA quality and quantity was done using a Bioanalyzer (Agilent, Santa Clara, CA, USA).

### RNA sequencing using the direct cDNA sequencing method

We used the Direct cDNA Sequencing Kit (SQK-DCS109, Oxford Nanopore Technologies, Oxford, United Kingdom) for the library preparation, following the manufacturer's instructions. One aliquot of RNA from each *Leptospira* strain was processed with the addition of poly(A) tail using the *Escherichia coli* poly(A) Polymerase Kit (New England Biolabs, Massachusetts, USA) following the protocols from the manufacturer. Before loading the samples in the flow cells, the polyadenylated and nonpolyadenylated samples underwent cDNA strand synthesis and switching processes, barcoding for multiplex sequencing, and adapter ligation. Up to six samples were multiplexed in a single run. The sequencing process was conducted on the MinION Sequencing platform (Oxford Nanopore Technologies) for 48 h with a bias voltage set to −180 mV.

### RNA sequencing using the direct RNA sequencing method

We sequenced RNA from two replicates of the LIC strain without the polyadenylation step using the Direct RNA Sequencing Kit (SQK-RNA002, Oxford Nanopore Technologies). The library preparation was done following the manufacturer's instructions. The library preparation process included attaching the RNA to the RT adapter and the sequencing adapters to the ends of the RNA. Since the Direct RNA Sequencing Kit does not allow multiplexing, each sequencing reaction was carried out in a single MinION flow cell for 24 h with a bias voltage set to −180 mV.

### Raw sequence quality check, basecalling, and demultiplexing

The quality of the raw cDNA and the DRS Fast5 files was first assessed using MinIONQC v.1.4.1 (36). After quality check, the sequences were base-called using Guppy v. 4.4.2 (https://community.nanoporetech.com) with the option "—config dna_r9.4.1_450bps_hac.cfg" for the cDNA samples and "—config rna_r9.4.1_70bps_hac.cfg" for the DRS samples in the FASTQ format. FASTQ files of cDNA reads were further demultiplexed using Guppy's "guppy_barcoder" function with the option "—barcode_kits EXP-NBD104." After demultiplexing, the barcoding summary produced with guppy_barcoder was used to separate each sample of multiplexed cDNA Fast5 files into the demultiplexed Fast5 files using the "demux_fast5" function provided by ont_fast5_api (https://github.com/nanoporetech/ont_fast5_api). The separated Fast5 files for each individual cDNA sample were then re-basecalled with the same configuration as above. MinIONQC was used again to assess the quality of each individual cDNA sample.

### Full-length reads identification and processing

Full-length (FL) reads from the cDNA raw sequence reads were identified using Pychopper v. 2.7.1 (https://github.com/epi2me-labs/pychopper) following the FL

identification protocol designed for bacterial RNA-Seq (33). We extracted the FL cDNA reads from each sample by determining the presence of the forward strand-switching primer (SSP; sequence: TTTCTGTTGGTGCTGATATTGCTGGG) and reverse anchored oligo(dT) VN primer (VNP; sequence: ACTTGCCTGTCGCTCTATCTTCTTTTTTTTT) within the individual raw reads using Pychopper. Reads that failed to identify the presence of both primers were considered read fragments and were filtered out. Subsequences identified outside of the SSP and VNP primers were trimmed from the raw reads along with the identified primers to obtain a FL read data set. The raw reads were first processed with the default Pychopper parameters to autotune the cutoff values from the cDNA read input of each sample. The reads that could not be identified as FL in the first round (unclassified reads) went through a rescue process using the parameter (-x rescue) in the second round of processing with a direct cDNA-specific rescue option (DCS109) turned on. The processed and rescued reads from both rounds were concatenated to create the FL read data set for each cDNA sample.

The average mapping qualities of the SSP and VNP primers identified from each raw cDNA read were obtained from the Pychoppers' summary reports (obtained with the "-A" option specified) (37). The length of the trimmed-off subsequences (which were subsequences trimmed from each end of the raw cDNA reads until the positions of the identified SSP and VNP primers, respectively) was obtained by comparing the original read length to the read-trimmed positions reported in the Pychoppers' summary report using a custom R script ("Rscript/get_trimmed_length.r" in the github repository https://github.com/rx32940/Lepto_transcriptome_ONT_cDNA).

## Reference genome mapping

Raw and FL cDNA reads were mapped to their corresponding reference genomes using Minimap2 v. 2.17 (38), separately, with the options "-ax splice -p 0.99 –MD –cs -Y." DRS reads were mapped with the options "-ax splice -uf -k14 -p 0.99 –MD –cs -Y." Strain-specific reference genomes were used for mapping, where *L. interrogans* serovar Copenhageni str. Fiocruz L1-130 (GCF_000007685.1) was used for LIC samples, *L. interrogans* serovar Icterohaemorrhagiae str. Langkawi (GCF_014858895.1) was used for LII samples, and *L. biflexa* serovar Patoc str. "Patoc 1 (Paris)" (GCF_000017685.1) was used for LBP samples. Each sample's alignment files in binary format were first sorted and indexed using SAMTools v.1.10 (39) and then converted into BED format for downstream analysis using the "bamtobed" function in BEDTools v. 2.30.0 (40) with the "-cigar -tag NM" options. We further obtained the mapping identities of all reads in each sample using a custom Python script with the formula ($1 - \frac{NM}{aligned read length} \times 100$) (33), where *NM* is the number of mismatches and gaps reported by Minimap2, and the *aligned read length* was reported as the sum of M(atch) and I(nsertion) characters obtained from each read alignment's CIGAR (Compact Idiosyncratic Gapped Alignment Report) string.

To determine the closest gene to each mapped read and the relative mapping positions of the read to the mapped gene, we used a custom Python script ("python_scripts /anot_read_transcript_v2.py" in the github repository https://github.com/rx32940/Lepto_transcriptome_ONT_cDNA). Here, we compared the mapping positions (mapping start and mapping end) of each read to the reference genome to the coordinates of the annotated genes provided by the corresponding reference genome's GFF file.

## Homopolymer of A regions evaluation

The method to assess the coverage of regions with homopolymers of A was adapted from a previous study (41). Genes with at least five consecutive A's were identified from all the reference genomes used in this study with a custom Python script ("python_scripts/isHomo_in_cDNA.py" in the github repository https://github.com/rx32940/Lepto_transcriptome_ONT_cDNA). The coverage of individual bases that were located 100 bp up- and downstream of the identified homopolymer of A regions was

obtained using the "coverage" function in BEDTools v. 2.30.0 (40) with the option "-d." For FL and DRS reads, the coverage of each base flanking the homopolymer of A regions was determined in a strand-specific manner with the option "-s." The relative coverage of each base flanking the homopolymer of A regions was calculated by using the absolute coverage of each base divided by the maximum coverage of the gene where the homopolymer was identified. The maximum coverage of each gene and the relative coverage of each base were determined by custom Python scripts ("python_scripts/trans_max_cov_pos.py" and "python_scripts/relative_cov_per_base.py" in the github repository https://github.com/rx32940/Lepto_transcriptome_ONT_cDNA).

## Poly(A) tail length estimation

We estimated the length of the poly(A) tail for both cDNA and DRS samples using two published tools: Nanopolish v 0.13.2 (42) and Tailfindr v 1.0 for DRS samples and Tailfindr v 1.0 for the cDNA sequenced samples (43).

## Unannotated genomic regions (new genes) identification

All the reads that were identified as mapping to the "noncoding" positions in the previous analysis were extracted and grouped together based on the closest annotated genes that they were mapped to. The groups with only one read were filtered out from the downstream analysis. After filtering, the mapped positions with the highest coverage at the 3′ and 5′ ends within each group of reads were obtained as the potential start and end positions of the unannotated genes, respectively. Nucleotide sequences of these new genes were extracted using the "getfasta" function in BEDTools v. 2.30.0 (40). The regions with duplicated sequences and sequences larger than 200,000 bp were filtered. The remaining sequences were queried in the Rfam database (44) for further annotation.

## Operons identification

Reads that were identified to cover more than one annotated gene were identified as operon reads. To avoid identifying operons from reads that are only partially covering a gene, only reads covering at least 90% of their mapped genes' annotated coordinates were used for operon identification. In the end, the transcriptional units, including the same annotated genes (based on each gene's annotated gene name), were collapsed together to identify the unique operons.

## Correlation between gene coverages

The number of reads mapped to each annotated gene was obtained using HTSeq v. 0.9.1 (45) with the option "—nonunique all –type = CDS." This option accounts for all reads, even the ones that were mapped to more than one gene, contributing to the coverage of the mapped gene to which the reads belong. The number of reads mapped to each gene was normalized into transcript per million (TPM) using a custom R code ("Read HTSeq" section in the "Rscript/all_read_stats.Rmd" file in the github repository https://github.com/rx32940/Lepto_transcriptome_ONT_cDNA). The correlation analyses between the normalized gene coverages were performed and determined using the "lm" function in the R "stats" package (37).

## RESULTS

### Overall sequencing and mapping results

A schematic representation of the sequencing process is shown in Fig. S1. The total RNA from the two pathogenic *L. interrogans* strains LIC and LII and one nonpathogenic strain of LBP was sequenced with ONT's cDNA protocol with and without additional synthetic polyadenylation. Almost 3 million reads with an average read length of ~1,764 bp were sequenced from the six *Leptospira* cDNA samples used in this study. Over 95% of the

sequenced cDNA reads in all samples had sequencing quality scores higher than seven (QUAL ≥7). The average read length was higher in samples [1,835 (SD: 91) bp long] with polyadenylation [poly(A)] than in samples [1,638 (SD: 115) bp long] without polyadenylation [nonpoly(A)]. The summary statistics of the cDNA sequences are shown in Table 1.

The FL cDNA reads consisted, on average, of ~10% of reads in the raw nonpolyadenylated samples and 50% of the reads in the raw polyadenylated samples (Table 1). After filtering and trimming, the average read length for the FL nonpolyadenylated samples decreased by almost two-thirds compared to the raw nonpolyadenylated samples [~566 (SD: 162) bp long]. In contrast, the average read length for the polyadenylated samples was only decreased by 12% [1,606 (SD: 81) bp long] (Table 1). In addition, the reverse primers, VNP, identified in the nonpolyadenylated cDNA reads had lower mapping quality and longer trimmed-off length (see Materials and Methods) during the FL read preprocessing step than those in the polyadenylated ones (Fig. S2).

A larger number of raw cDNA reads (>99%) were mapped to the corresponding reference genomes for both polyadenylated and nonpolyadenylated samples than the FL cDNA reads (91%) (Fig. 1a). The majority of the raw (94%) and FL (95%) cDNA reads in each sample were mapped to the rRNA genes. Only 5% and 3% of the raw and FL reads, respectively, were mapped to the mRNA genes (Fig. 1b). A higher mapping rate to the mRNA genes was observed in both raw and FL cDNA reads from the polyadenylated samples than from the nonpolyadenylated ones. A higher number of annotated genes were mapped at least once in the polyadenylated cDNA samples than in the nonpolyadenylated ones (Fig. 1c).

To further evaluate the presence and characteristics of the nonpolyadenylated RNA fraction, we proceeded with ONT's DRS sequencing protocol to sequence the original nonpolyadenylated RNA molecules without reverse transcription on two technical replicates of the LIC strain (LIC-DRS). The DRS samples showed a relatively high sequencing yield (Table 2), with an average read length of 767 and 868 bp from each DRS sample. Over 90% of the sequenced reads had a quality score of over seven (Table 2). Our attempt to sequence RNA by DRS after the polyadenylation step generated only less than ~0.025 Gb of data, and these were not enough for a comparison analysis (data not shown).

An average of 36% of DRS reads were mapped to the reference genome (Fig. 2a), with an average mapping identity lower (87%) than that of cDNA reads (raw: 90%; FL: 93%). In contrast to the cDNA reads, over 71% of the DRS reads were mapped to the mRNA genes, and only 28% of the reads were mapped to the rRNA genes (Fig. 2b). The DRS reads were sequenced without polyadenylation and aligned to more distinct annotated genes (Fig. 2c) than the cDNA polyadenylated LIC sample (Fig. 1c).

The relative mapping positions of each direct cDNA or DRS read in relation to the coordinates of the closest annotated genes it was mapped to (in its corresponding reference genome) are shown in Fig. S3. This figure also shows the composition of the transcript fragments within each sample.

**TABLE 1**   Quality statistics of the *Leptospira* raw cDNA reads

| Sample ID | Raw read statistics | | | | | | | FL read statistics | |
|---|---|---|---|---|---|---|---|---|---|
| | Number reads | Read QUAL ≥7 | Cumulative size (Gb) | Mean read length (bp) | N50 (bp) | Max read length (bp) | Number reads >1 kb | Number reads | Mean read length (bp) |
| LIC-cDNA-nonpolyA | 199,615 | 184,279 | 0.32 | 1,601 | 2,230 | 24,092 | 139,679 | 29,821 | 520 |
| LIC-cDNA-polyA | 647,339 | 611,858 | 1.26 | 1,939 | 2,987 | 31,992 | 464,560 | 365,534 | 1,664 |
| LII-cDNA-nonpolyA | 317,151 | 292,752 | 0.49 | 1,547 | 2,167 | 17,330 | 214,681 | 41,000 | 433 |
| LII-cDNA-polyA | 512,875 | 480,989 | 0.92 | 1,799 | 2,841 | 47,547 | 351,270 | 249,622 | 1,514 |
| LBP-cDNA-nonpolyA | 516,304 | 503,805 | 1 | 1,928 | 2,877 | 20,570 | 420,667 | 40,006 | 746 |
| LBP-cDNA-polyA | 703,577 | 680,593 | 1.24 | 1,768 | 2,556 | 30,998 | 497,475 | 312,858 | 1,640 |

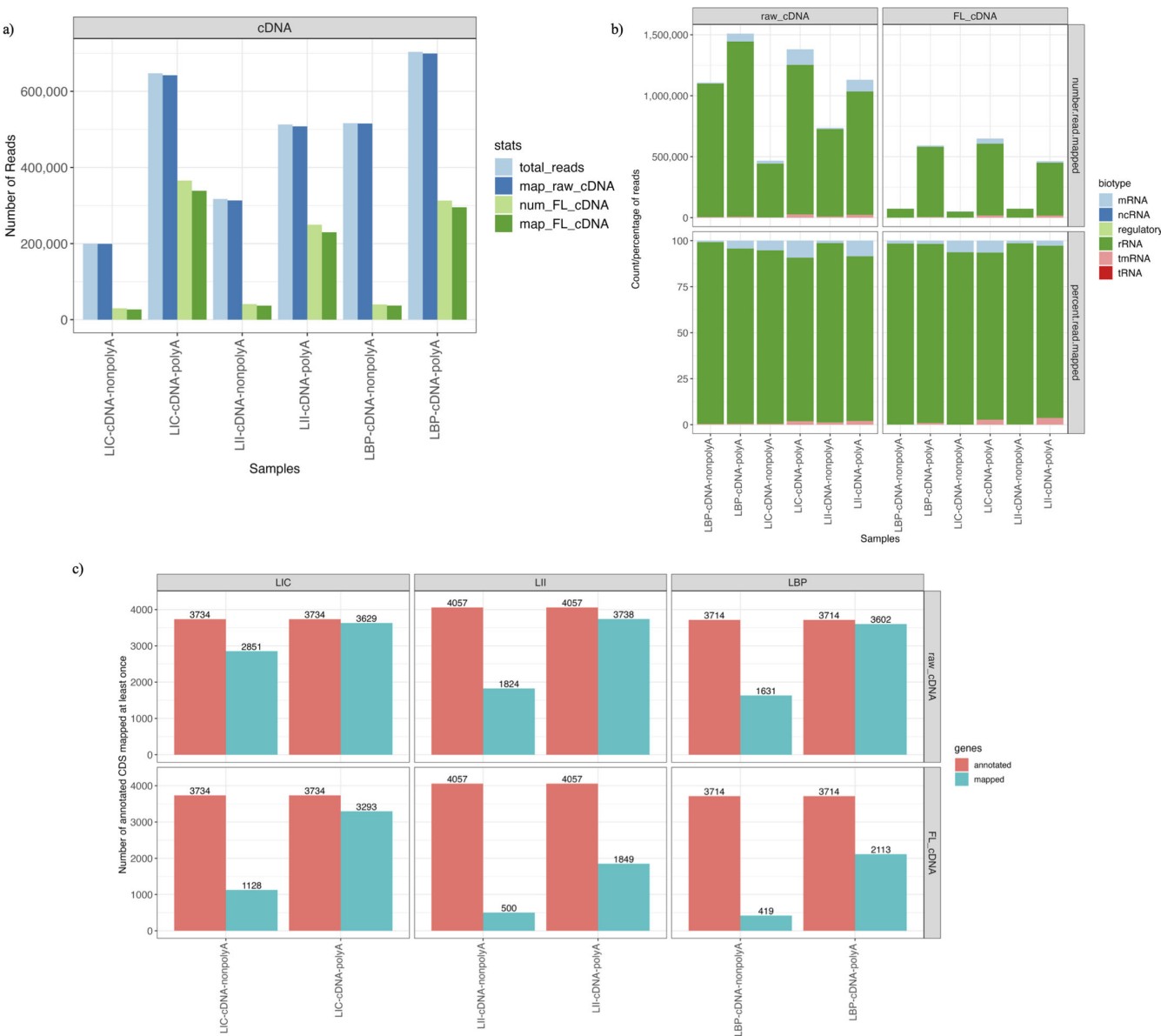

**FIG 1** Mapping statistics of the cDNA reads. (a) The total number of raw (light blue) and FL (light green) cDNA reads sequenced within each sample and the number of raw (dark blue) and FL (dark green) reads mapped to their corresponding reference genomes. (b) The number (top) and percentage (bottom) of reads mapped to different RNA biotypes for raw (left) and FL cDNA reads (right). (c) The number of unique genes mapped by raw cDNA (top row) and FL cDNA (bottom row). Each column represents the samples obtained from a different strain (LIC, LII, and LBP). The red bar on the left is the total number of annotated genes in the corresponding reference genomes, and the blue bar on the right is the number of unique genes mapped at least once in the sample.

## Gene coverages

To compare the RNA profiles of the cDNA reads with and without polyadenylation, we calculated for each sample the correlation between their corresponding mapped gene coverages. The gene coverage correlations between nonpolyadenylated and polyadenylated samples for the raw cDNA reads showed large variation across strains. The correlation values ranged from $R^2 = 0.478$ (LII samples) to $R^2 = 0.704$ (LIC samples) (Fig. 3a). The correlation values between the nonpolyadenylated and polyadenylated samples evaluated with cDNA FL reads showed slightly lower values for LIC ($R^2 = 0.549$) and LBP ($R^2 = 0.476$), except for LII ($R^2 = 0.524$) (Fig. S4).

**TABLE 2** Raw DRS reads quality statistics

| | Raw read statistics | | | | | | |
|---|---|---|---|---|---|---|---|
| Sample ID | Number reads | Read QUAL ≥7 | Cumulative size (Gb) | Mean read length (bp) | N50 (bp) | Max read length (bp) | Number reads >1 kb |
| **LIC-DRS-R1** | 1,622,692 | 1,482,900 | 1.25 | 767 | 1,229 | 48,619 | 586,993 |
| **LIC-DRS-R2** | 483,847 | 482,807 | 0.42 | 868 | 1,320 | 6,915 | 219,718 |

We further examined the correlation between the annotated gene coverages of the two sequencing protocols, cDNA and DRS (Fig. 3b), of the LIC samples. The correlation between the LIC-DRS reads and each of the polyadenylated and nonpolyadenylated LIC cDNA reads showed a medium correlation between the reads from the two sequencing methods, where the correlations between the LIC-DRS reads were highly correlated with each other ($R^2$ ~0.93). LIC-DRS reads' correlations with the polyadenylated LIC cDNA reads were slightly higher ($R^2$ ~0.66) than their correlations with the nonpolyadenylated LIC cDNA reads ($R^2$ ~0.56).

To evaluate the transcripts of the annotated genes that underwent potential posttranscriptional polyadenylation, we identified the top 50 most sequenced transcripts across the different strains (Table S1). Many of these highly sequenced transcripts were identified from both cDNA (nonpolyadenylated and polyadenylated) samples and from DRS reads. Some of these transcripts encode for flagellin/flagellar structural proteins, response regulators, ribosomal subunits, RNA transcription factors, catabolism pathway-related proteins, and surface lipoproteins. A total of 25 out of the 50 identified transcripts present in all the LIC samples were sequenced by the two different ONT sequencing protocols.

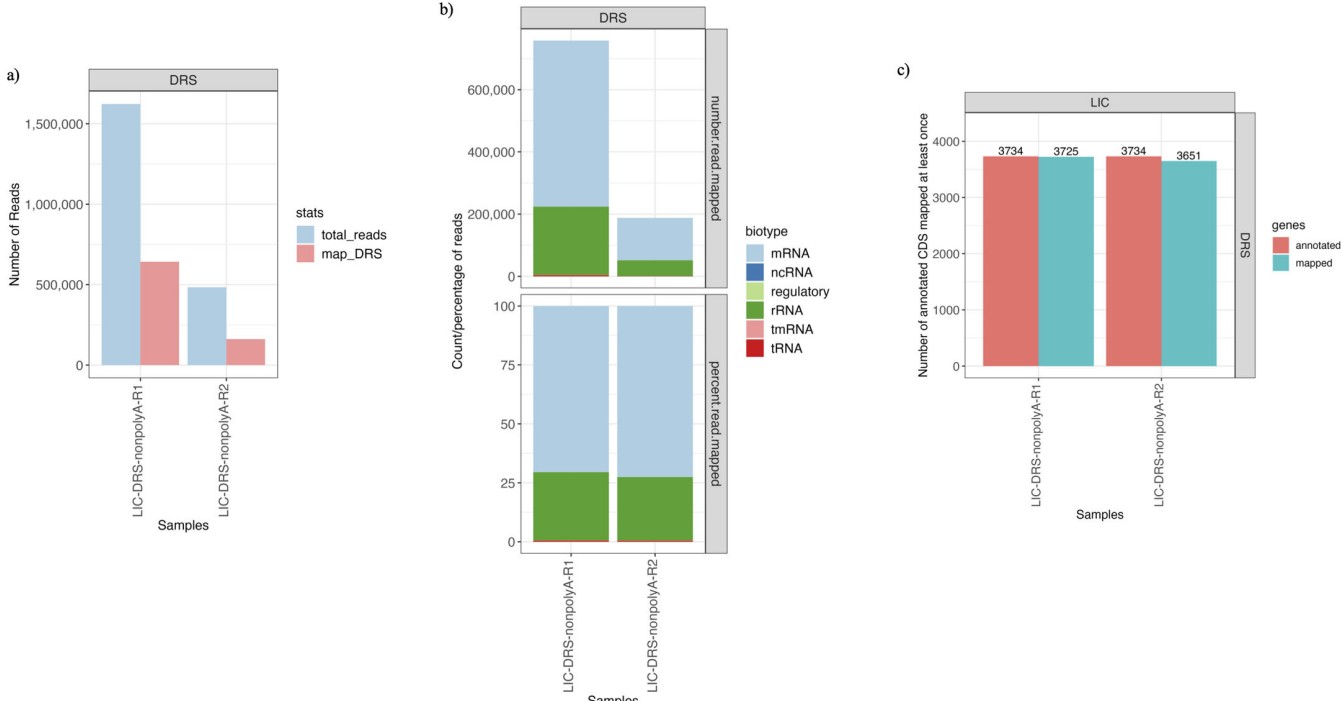

**FIG 2** DRS samples' reads and mapping statistics. (a) The number of DRS reads (light blue) sequenced within each sample and the number of DRS reads mapped to LIC's reference genome (light red). (b) The number (top) and percentage (bottom) of reads mapped to different RNA biotypes within each DRS sample. (c) The number of distinct genes mapped by DRS reads within each sample. The red bar on the left is the total number of annotated genes in the corresponding reference genomes, and the blue bar on the right is the number of distinct genes mapped at least once in the sample.

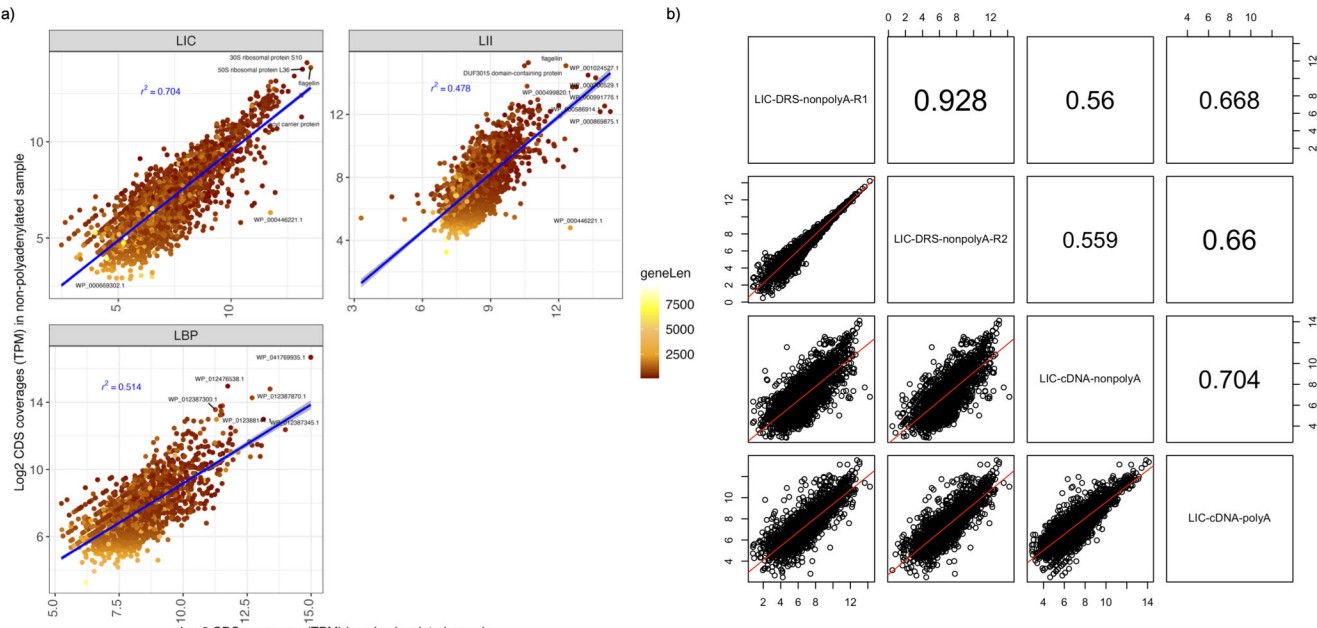

**FIG 3** Correlation between coverages of mapped genes. Correlations of mapped gene coverages (in $\log_2$ of TPM) between polyadenylated and nonpolyadenylated raw sequenced cDNA samples and between the LIC samples sequenced with the cDNA and DRS protocols. (a) Correlations of mapped gene coverages between the nonpolyadenylated cDNA (y-axis) and the polyadenylated cDNA (x-axis) samples. Each dot represents an annotated gene mapped by both the nonpolyadenylated and polyadenylated samples of each strain to their corresponding reference genome. The color of each dot is scaled by the length (in bp) of the corresponding genes. (b) Correlations of annotated gene coverages between the cDNA- and DRS-sequenced LIC samples. The numbers in the upper right diagonal boxes in the matrix are the $R^2$ values for the corresponding comparisons between samples labeled in the diagonal boxes directly below or left of the $R^2$ value.

## Identification of operons

Operons were identified as reads covering more than one annotated gene, and these reads accounted for less than 1% of the reads across all samples. In total, 791 unique operons were identified in this study. Only 45 of the identified operons mapped to more than five annotated genes, and most operons (550) covered exactly two annotated genes. Many of the identified operons aligned to genes with their annotated protein coding sequences on both leading and lagging strands of the reference genomes (Table S2).

Most of the operons identified in this study were uniquely identified from samples of a single strain (Fig. 4a). Only 15 operons were identified across all strains. Some operons (102) were shared only between the samples of the two pathogenic strains, LIC and LII, included in this study (Fig. 4a). Raw cDNA reads identified the largest number of unique operons, followed by the number of the DRS and FL cDNA ones (Fig. 4b). Almost all the operons identified in the FL cDNA samples were also identified in the raw cDNA samples. More than half of the operons identified from samples of raw cDNA and the DRS data sets overlapped with each other (Fig. 4b).

## Identification of unannotated genomic regions

A total of 1,113 unique unannotated genomic regions were identified across all samples (Table S3). After further querying sequences of these unannotated regions in the RNA database Rfam, we found that only a small portion of these sequences was previously described in *Leptospira* or in other bacteria's annotated genomes (Table S3). In general, more unannotated regions were identified in the genomes of the pathogenic *Leptospira* (Fig. 5a). Most of these unannotated regions were observed in the polyadenylated cDNA sequencing and DRS samples, and only a small number of these regions were observed

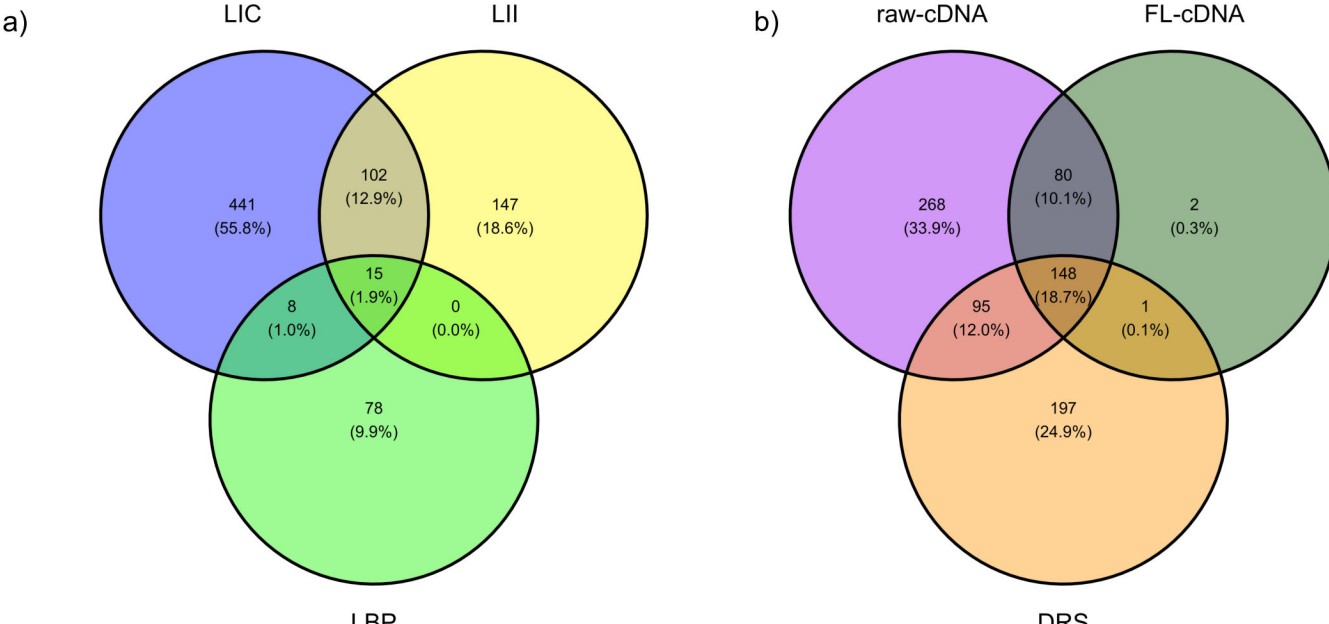

**FIG 4** Operon reads identified with DRS and raw cDNA reads. (a) Venn diagram showing the number of overlapping operon structures identified from the different *Leptospiral* strains in this study (LIC, LII, LBP). (b) Venn diagram showing the number of unique operon structures identified and shared by different data sets in this study (cDNA, direct cDNA sequencing; FL).

in the nonpolyadenylated cDNA samples (Fig. 5a). In addition, we found that more than half of these unannotated regions were syntenic between the two pathogenic strains (Fig. 5b).

### Evaluation of polyadenylated transcripts

The reads in the cDNA samples possess both poly(A) and poly(T) tails from the two consecutive reverse transcription steps. The polyadenylated samples had relatively longer median "A" and "T" stretches (~28 bp for A tails and ~57 bp for T tails) than the nonpolyadenylated ones (~16 bp for A tails and ~22 bp for T tails) (Fig. 6a). In the DRS protocol without polyadenylation, the native RNA molecules with native poly(A) tails are the only poly(A) stretches that had their lengths estimated. Only moderate correlation ($R^2 = 0.59$) was observed between the tail lengths estimated by the two popular DRS tail length estimation software programs, Nanopolish and Tailfindr. Nanopolish reported a median tail length of ~9 bp, and Tailfindr reported a median tail length of ~15 bp (Fig. 6b).

As depicted in Fig. 7a, ONT sequencing methods have the potential to prime and sequence the fragmented transcripts with truncations at the homopolymers of adenine (A) regions within the transcribed gene sequences. Sequencing enrichment of transcripts truncated at the homopolymers of the adenine (A) region can be reflected by a coverage drop in an alignment after the presence of the homopolymers of the adenine (A) region on the reference genome. To assess the bias introduced due to transcripts' truncation at the homopolymers of A region, we analyzed the average relative coverage (see Materials and Methods) of bases upstream and downstream of all identified homopolymers of A regions (position 0) from all the annotated genes (Fig. 7b).

Compared to the polyadenylated cDNA samples, the nonpolyadenylated raw cDNA samples show a slight drop in their average relative coverage after the presence of the homopolymer of A. This implies the biased enrichment in capturing fragmented transcripts truncated at homopolymers of A regions during library preparation with the absence of the synthetic poly(A) tails in the direct cDNA samples (Fig. 7b). However, the average coverage drop after the homopolymer of A regions was steeper within the

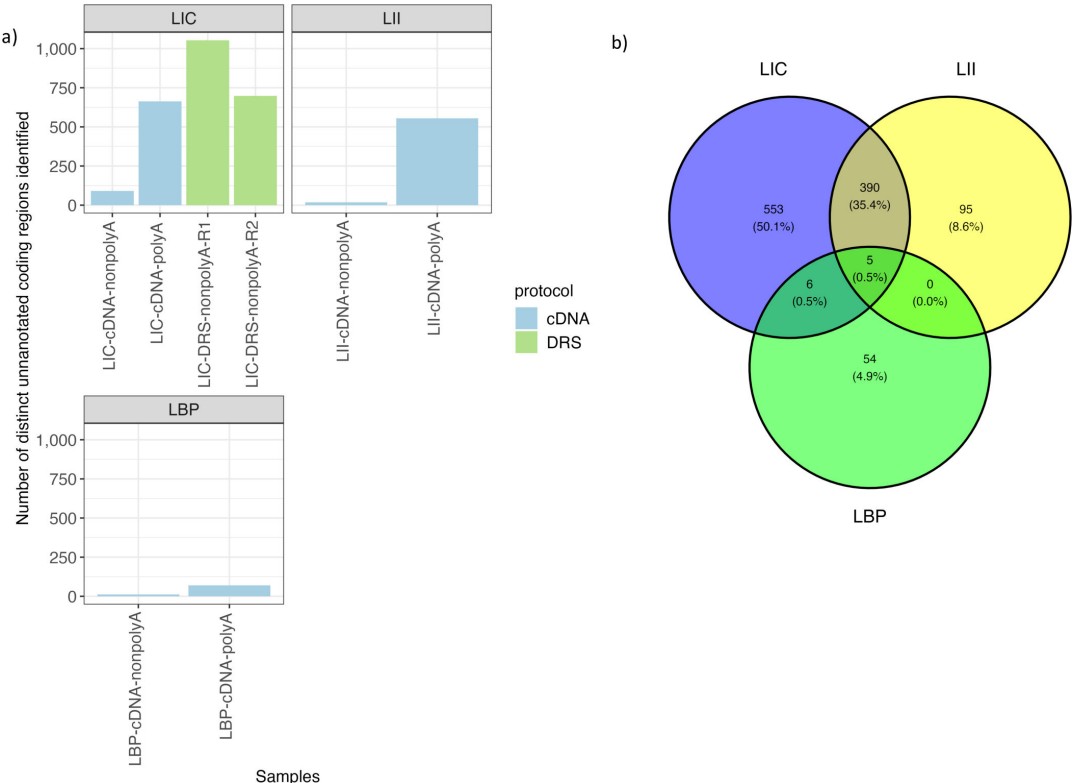

**FIG 5** Unannotated genomic regions identified in each *Leptospira* strain. (a) The number of unique unannotated coding regions identified from samples of the different strains (LIC, LII, LBP; cDNA, direct cDNA sequencing). (b) Venn diagram showing the number of intersected unannotated coding regions identified by the samples of different strains.

FL samples, especially in the nonpolyadenylated ones, implying that the FL processing algorithm truncated the nonpolyadenylated cDNA reads at their homopolymer of A regions during primer trimming in the absence of synthetically attached poly(A) tails. In addition, the average relative coverages flanking the homopolymers of A regions in

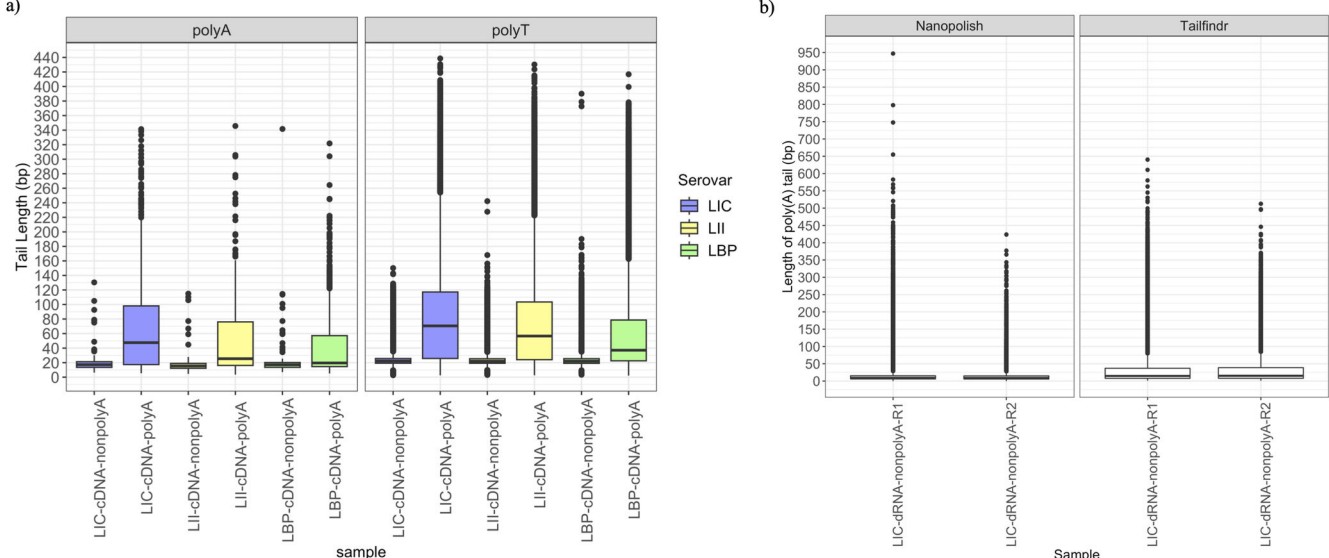

**FIG 6** Poly(A) tail assessment for direct cDNA and DRS reads. (a) The estimated tail length [poly(A) and poly(T)] for the direct cDNA samples. (b) The estimated tail lengths reported by two different software programs used to evaluate poly(A) tail length in DRS samples.

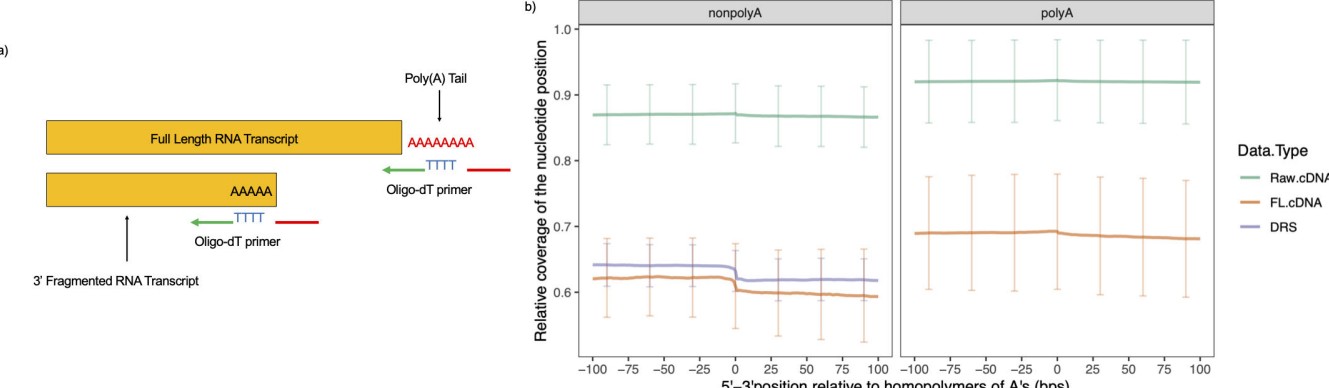

**FIG 7** Enrichment in sequencing transcripts with 3′ truncation at a homopolymer of A regions. (a) Oligo-dT primers in both cDNA and DRS protocols may pick up RNA transcripts fragmented at homopolymer of A regions within the sequences to initiate transcription during library preparation. (b) Average relative coverage for nucleotides flanking the homopolymer of A regions (≥5 bp) in the raw/FL cDNA and DRS samples. The relative coverage at each nucleotide position was calculated in relation to the maximum coverage within the annotated gene where the homopolymer was identified.

the LIC-DRS samples were largely dropped after the presence of the homopolymer of A regions (Fig. 7b), implying that DRS sequencing with no synthetic poly(A) tail added was enriched for the sequencing of the transcript fragments truncated at the homopolymer of A regions.

## DISCUSSION

In this study, we utilized ONT's long-read sequencing technology to evaluate the *Leptospira* transcriptome, identify potential new genes that have not been previously described, and determine operons encoding for transcripts of multiple genes. Our analysis suggested that some of the sequences containing homopolymers of adenine bases at the 3′ end of RNA molecules may, in fact, be the result of posttranscriptional polyadenylation described in prokaryotes (28).

For the unannotated genomic regions (new genes) identified in this study, most of them had never been reported previously in the *Leptospira* transcriptome. Many of the relative positions of these genes were consistent in the identities of their neighboring genes across the reference genomes of the pathogenic *Leptospira* strains analyzed in this study and could be the result of large horizontal gene transfer events with the involvement of multiple genes during leptospiral evolution (46). Furthermore, most of the new genes were not detected in the nonpathogenic *Leptospira*, suggesting that the RNA molecules transcribed from these new genes may play potential pathogenesis-related roles during adaptation. More interestingly, many operons identified in the pathogenic *Leptospira* were also not observed in the nonpathogenic *Leptospira*, suggesting different transcriptional mechanisms present in the pathogenic and nonpathogenic *Leptospira* species. Carefully designed experiments are needed to unravel the role of these new genes and transcriptional units in leptospiral pathogenesis. Although we observed some evidence for variations in RNA expression between strains analyzed in this study, the data should be interpreted with caution considering (i) the inherent issues associated with the ONT sequencing methods, (ii) the lack of technical replicates (32, 33, 47, 48), and (iii) the differences in leptospiral transcriptional activities and pathogenesis mechanisms *in vitro* vs *in vivo* (49). We propose further confirmation of these findings using hybrid sequencing and assembly methods. Additionally, we recommend carefully designed experiments with appropriate replicates to examine differences in leptospiral transcriptional activities between pathogenic and nonpathogenic strains within some of their common hosts (e.g., rodents).

The ONT's direct long-read sequencing protocols sequence the full-length single-molecule RNA transcript per read with no theoretical read-length limitations or amplification

biases (50), thus allowing the detection of different isoforms and intermediate structures of RNA transcripts (35, 48). In addition, ONT's direct sequencing of RNA molecules can assess the presence of posttranscriptional modification structures in the sequenced libraries, including the presence of poly(A) tails (32, 51). In our initial experiments with ONT's RNA-Seq technologies, despite skipping the polyadenylation step required for sequencing prokaryotic mRNA, the observation of obtaining a moderate yield of transcript reads was both interesting and intriguing. Subsequent comparisons were made using pathogenic and nonpathogenic *Leptospira* using RNA transcripts sequenced with and without polyadenylation. While there were no differences in quality or read length between nonpolyadenylated and polyadenylated reads, the sequencing yield in the nonpolyadenylated direct cDNA samples and their percentage of reads identified as FL reads were significantly smaller than those of the polyadenylated direct cDNA samples. This is expected because only transcripts with a native poly(A) tail attached were expected to get sequenced in the nonpolyadenylated libraries. In comparison to both the nonpolyadenylated and polyadenylated direct cDNA samples, the nonpolyadenylated DRS samples contained a higher percentage of mRNA molecules that were sequenced from a higher number of distinct annotated genes. However, the average read time was shorter than both the nonpolyadenylated and polyadenylated direct cDNA samples. This could be due to the DRS's capability of capturing more intermediate RNA transcripts in the process of degradation (52). In addition, the nonpolyadenylated DRS-sequenced samples had a lower percentage of their sequenced reads mapped to the reference genome than the cDNA-sequenced ones. This suggests that DRS reads suffered from a higher error rate during sequencing (32), which was further confirmed by the lower mapping identity evaluated from the DRS samples' alignments. In the end, our attempt to sequence RNA by DRS with the synthetic polyadenylation step generated only less than ~0.025 Gb of data, and these were not enough for a comparative analysis. We believe that this could be a result of pore blocking by the RNA secondary structures during the sequencing process (33, 47).

Although largely recognized and described in eukaryotes, the presence and function of polyadenylation resulting in 3′ poly(A) tails in the prokaryotic transcriptome have not been well characterized (6, 8, 20, 28, 53). Previous studies of prokaryotic polyadenylation have found that, unlike the eukaryotic poly(A) tails, which are mainly attached to the 3′ end of mRNA transcripts (and some long noncoding RNAs and small RNAs) (30, 54), the polyadenylation sites in the prokaryotic transcriptome can attach to mRNA, rRNA, tRNA, tmRNA, and sRNA molecules (55). Due to their role in RNA degradation mechanisms, prokaryotic poly(A) tails can be identified not only at the end of full-length primary transcripts but also in the cleaved intermediate transcripts processed by endonucleases (56, 57). The presence of a poly(A) polymerase (PAP), which is the enzyme attaching adenine bases to the 3′ end of a transcript after transcription, was described in *E. coli* as early as 1962, and the gene encoding for PAP, *pcnB,* was also identified in *E. coli* (31, 58). Later, this gene was identified in a limited number of Gram-negative bacterial genomes (29, 59). Interestingly, the differential expression of the *pcnB* gene has been identified in a previous transcriptome study of the pathogenic *Leptospira* species, *L. interrogans*, when cultured under different conditions (6). We also identified that many of the highly sequenced transcripts in the nonpolyadenylated samples are consistent across strains and species analyzed, suggesting the presence of shared mRNA profiles potentially regulated by inherent polyadenylation across *Leptospira* species. It is noteworthy that many of these highly sequenced transcripts in the nonpolyadenylated samples were also found to be regulated by the polyadenylation in *E. coli*. These include transcripts encoding for membrane proteins, motility proteins, and RNA metabolism proteins (28, 60). However, previous phylogenetic analysis showed that the *pcnB* gene of *Leptospira* has a different origin from that of *E. coli* and other proteobacteria, thus *Leptospira* poly(A) tail might be regulating *Leptospira* RNA levels under different mechanisms (29).

Although our study provided the opportunity to explore leptospiral transcriptomic landscapes and RNA structures without synthetic polyadenylating samples, it also

presented a few limitations. In addition to sequencing transcripts that possess a native poly(A) tail, transcripts truncated at the homopolymers of A regions during library preparation were captured, especially in the nonpolyadenylated DRS samples, which can be a significant proportion of RNA profiles from the AT-rich *Leptospira* genomes. While this method is advantageous for studying posttranscriptional polyadenylation, the sequencing of transcripts truncated at the homopolymer of A regions could be the source of bias for making conclusions in RNA composition and differential gene expression studies. We also observed that the algorithms (Pychopper) used to identify FL cDNA reads could falsely identify sequences in the middle of a raw cDNA read with similar nucleotide identity as the reverse primers [which contain poly(T) stretches capturing the poly(A) stretches of targeted transcripts; see Materials and Methods]. This misidentification of reverse primer will falsely classify a nonFL read as a FL cDNA read and trim a raw cDNA read in the middle of the sequenced transcripts from the misidentified primer, especially in the nonpolyadenylated cDNA samples, leading to the enrichment of homopolymer of A truncated reads after processing. Thus, DRS and the FL cDNA read identification process should be taken with caution when sequencing prokaryotic transcriptomes without polyadenylation.

These biases may influence the conclusion about the presence of poly(A) tails in the prokaryotic transcriptome and the relative profiles of gene expressions. However, the range of poly(A) tail length estimated from the nonpolyadenylated cDNA samples (~16 bp) and one of the tail length estimation software for the DRS samples (~15 bp using Tailfindr) were comparable with the previous estimation for prokaryotic poly(A) tail *in E.* coli (14–60 bp) (28). Among the software we used for poly(A) tail length estimation, only one software is currently developed for direct cDNA tail length estimation, and therefore, in this study, it was not possible to compare the accuracy of poly(A) tail estimation in the direct cDNA samples. In addition, a recent study has reported that the use of ONT direct cDNA for tail length estimation might report erroneous information due to the possibility of the oligo(T) primer of direct cDNA sequencing attaching to the middle of a poly(A) tail to initiate the reverse transcription reaction, thus only capturing a part of the poly(A) tail length of the targeted RNA transcript (61). We also observed that the results from the two poly(A) tail length estimation software used for DRS samples were not consistent with each other. Therefore, we should be cautious about tail length estimations with current technologies, and improvements in the methods and/or further experimental verification are needed to assess this feature.

Finally, our study focused on comparing the transcriptional profiles of pathogenic and nonpathogenic *Leptospira* strains cultured *in vitro*; however, these results may not reflect within-host leptospiral pathogenicity (49), and we suggest that our findings should be further validated *in vivo*.

## Conclusion

We present a preliminary comparison of *Leptospira* transcriptomes using the ONT's direct cDNA and RNA sequencing protocols. ONT's ability to detect and sequence prokaryotic RNA molecules without polyadenylation opens an opportunity to explore potential posttranscriptional polyadenylation and its effects on prokaryotic gene regulation. Further evaluation of the identified new genes and operons discovered in this study will unravel the complex virulence, pathogenesis, and host adaptation mechanisms of *Leptospira*, an understudied spirochete of public health significance.

## ACKNOWLEDGMENTS

We thank the Interdisciplinary Disease Ecology Across Scales program (funded by the National Science Foundation under Grant No. DGE-1545433) at the University of Georgia for providing funding and training to R.X. when completing this manuscript; the University of Tennessee, College of Veterinary Medicine, for start-up funds for S.R. to set

up the *Leptospira* research laboratory; and the University of Georgia, Office of Research, for start-up funds for L.C.M.S.

## AUTHOR AFFILIATIONS

[1]Institute of Bioinformatics, University of Georgia, Athens, Georgia, USA
[2]Center for the Ecology of Infectious Diseases, University of Georgia, Athens, Georgia, USA
[3]Department of Biomedical and Diagnostic Sciences, College of Veterinary Medicine, University of Tennessee, Knoxville, Tennessee, USA
[4]Department of Infectious Diseases, College of Veterinary Medicine, University of Georgia, Athens, Georgia, USA

## PRESENT ADDRESS

Dhani Prakoso, Professor Nidom Foundation, Surabaya, Indonesia
Liliana C. M. Salvador, School of Animal and Comparative Biomedical Sciences, University of Arizona, Tucson, Arizona, USA

## AUTHOR ORCIDs

Ruijie Xu  http://orcid.org/0000-0003-3364-1587
Liliana C. M. Salvador  http://orcid.org/0000-0001-8472-1179
Sreekumari Rajeev  http://orcid.org/0000-0001-6600-9102

## AUTHOR CONTRIBUTIONS

Ruijie Xu, Data curation, Formal analysis, Methodology, Software, Validation, Visualization, Writing – original draft, Writing – review and editing | Dhani Prakoso, Data curation, Investigation, Methodology, Writing – review and editing | Liliana C. M. Salvador, Formal analysis, Funding acquisition, Methodology, Project administration, Resources, Software, Supervision, Validation, Visualization, Writing – original draft, Writing – review and editing | Sreekumari Rajeev, Conceptualization, Funding acquisition, Investigation, Methodology, Project administration, Resources, Supervision, Visualization, Writing – original draft, Writing – review and editing

## DATA AVAILABILITY

All direct cDNA sequenced samples are available under bioproject: PRJNA949181, with biosample accession: SAMN33927558, SAMN33927559, SAMN33927560, SAMN33927561, SAMN33927564, SAMN33927565. All direct RNA sequenced samples are available under bioproject: PRJNA949189, with biosample accession: SAMN33933649 and SAMN33933650. All scripts for mapping analysis are available in the github repository: https://github.com/rx32940/Lepto_transcriptome_ONT_cDNA.

## ADDITIONAL FILES

The following material is available online.

### Supplemental Material

**Supplemental figures (Spectrum02234-23-s0001.pdf).** Fig. S1, S2, S3, S4.
**Table S1 (Spectrum02234-23-s0002.xlsx).** Top 50 most highly mapped annotated coding regions (TPM) mapped by both the nonpolyadenylated and polyadenylated samples in each serovar outside of 16S and 23S rRNA coding regions.
**Table S2 (Spectrum02234-23-s0003.xlsx).** Operon structures identified from samples of each data set.

**Table S3 (Spectrum02234-23-s0004.xlsx).** Unannotated genomic regions identified in each sample.

## Open Peer Review

**PEER REVIEW HISTORY (review-history.pdf).** An accounting of the reviewer's comments and feedback.

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
