## [Reviewer comments · Microbiology Spectrum]

Microbiology Spectrum

Leptospira transcriptome sequencing using long-read technology reveals unannotated transcripts and potential polyadenylation of RNA molecules

Ruijie Xu, Dhani Prakoso, Liliana Salvador, and Sreekumari Rajeev

Corresponding Author(s): Sreekumari Rajeev, UT AgResearch

Review Timeline:

Submission Date:	May 26, 2023
Editorial Decision:	August 14, 2023
Revision Received:	August 30, 2023
Accepted:	September 11, 2023

Editor: Jonathan Jacobs

Reviewer(s): Disclosure of reviewer identity is with reference to reviewer comments included in decision letter(s). The following individuals involved in review of your submission have agreed to reveal their identity: Feng Xue (Reviewer #2)

Transaction Report:

DOI: <https://doi.org/10.1128/spectrum.02234-23>

August 14, 2023

Dr. Sreekumari Rajeev
UT AgResearch
Biomedical and Diagnostic Sciences
2407 River Dr
Knoxville 37996

Re: Spectrum02234-23 (Leptospira transcriptome sequencing using long-read technology reveals unannotated transcripts and potential polyadenylation of RNA molecules)

Dear Dr. Sreekumari Rajeev:

Thank for your patience while we waited for responses from the reviewers. Personally, this was an excellent manuscript and well written. I personally did research on mRNA degradation and translational control of gene expression as part of my graduate work, so this paper was fun to read and feel it's an important part of the story of polyA in prokaryotes. That being said, I would like for you to respond to the reviewers comments and update the manuscript accordingly. Please send me the revised manuscript once it's completed. I do not plan to send it back out for peer review, but will review it myself prior to making a final decision.

Link Not Available

Sincerely,

Jonathan Jacobs

Journals Department
Reviewer comments:

Reviewer #1 (Public repository details (Required)):

The authors performed RNA sequencing and have uploaded the results to a public database.

Reviewer #1 (Comments for the Author):

Summary

In this article, Xu et al. sequenced and analyzed the transcriptome of 2 pathogenic strains and 1 nonpathogenic strain of *Leptospira* through Oxford Nanopore Technologies' direct cDNA and direct RNA sequencing methods. They identified new operons, RNA molecules, and evidence of potential posttranscriptional polyadenylation in *Leptospira* transcriptomes. The use of ONT's sequencing may improve the understanding of leptospiral RNA landscapes and polyadenylation. Here, I have the several questions and comments.

Comments

1. In some parts of the text, the writing of 'leptospira's' should be replaced with 'leptospiral', such as line 36, line 59, and line 60.
2. line 72, a spelling mistake of 'INDRODUCTION' should be corrected.
3. line 126, just a word of 'bacteria' is not enough to summarize the follow up contents of the method.
4. line 131, 'a 4-day old culture' should be changed with 'a 4-day culture'.
5. line 127-line137, some important details are missed. Passage times of pathogenic *Leptospira*? Because the virulence of the pathogenic *Leptospira* decreases rapidly during in vitro passage. The number of leptospire used for RNA extraction.
6. The differential RNA expression detected between pathogenic and nonpathogenic leptospire should be further validated by qPCR.
7. In addition to posttranscriptional polyadenylation modification, is there any other posttranscriptional modifications could be detected by ONT's RNA-Seq technologies?
8. In the figure legend of Figure S4, 'IC: *L. interrogans*' should be corrected with 'LIC: *L. interrogans*'.

Reviewer #2 (Public repository details (Required)):

The raw data had been deposited in a NCBI-SRA.

Reviewer #2 (Comments for the Author):

1. The English grammar should be revised for publication.
2. This research provided a detailed transcriptome profiles of the pathogen *Leptospira*. However, the author did not emphasize their new finding in the main text figures.
3. The major defect of the research design is that the pathogen was in vitro cultured, not the separated culture from cell or animal models. So this transcriptome may not reveal the pathogenesis of the pathogen in vivo. The author should mentioned the limitation in Discussion with references.

Staff Comments:

Preparing Revision Guidelines

Please return the manuscript within 60 days; if you cannot complete the modification within this time period, please contact me. If you do not wish to modify the manuscript and prefer to submit it to another journal, please notify me of your decision immediately so that the manuscript may be formally withdrawn from consideration by Microbiology Spectrum.

Summary

In this article, Xu et al. sequenced and analyzed the transcriptome of 2 pathogenic strains and 1 nonpathogenic strain of *Leptospira* through Oxford Nanopore Technologies' direct cDNA and direct RNA sequencing methods. They identified new operons, RNA molecules, and evidence of potential posttranscriptional polyadenylation in *Leptospira* transcriptomes. The use of ONT's sequencing may improve the understanding of leptospiral RNA landscapes and polyadenylation. Here, I have the several questions and comments.

Comments

1. In some parts of the text, the writing of 'leptospira's' should be replaced with 'leptospiral', such as line 36, line 59, and line 60.
2. line 72, a spelling mistake of 'INDRODUCTION' should be corrected.
3. line 126, just a word of 'bacteria' is not enough to summarize the follow up contents of the method.
4. line 131, 'a 4-day old culture' should be changed with 'a 4-day culture'.
5. line 127-line137, some important details are missed. Passage times of pathogenic *Leptospira*? Because the virulence of the pathogenic *Leptospira* decreases rapidly during *in vitro* passage. The number of leptospire used for RNA extraction.
6. The differential RNA expression detected between pathogenic and nonpathogenic leptospire should be further validated by qPCR.
7. In addition to posttranscriptional polyadenylation modification, is there any other posttranscriptional modifications could be detected by ONT's RNA-Seq technologies?
8. In the figure legend of Figure S4, 'IC: *L. interrogans*' should be corrected with 'LIC: *L. interrogans*'.

Spectrum02234-23

Leptospira transcriptome sequencing using long-read technology reveals unannotated transcripts and potential polyadenylation of RNA molecules

Editor Comments:

Thank for your patience while we waited for responses from the reviewers. Personally, this was an excellent manuscript and well written. I personally did research on mRNA degradation and translational control of gene expression as part of my graduate work, so this paper was fun to read and feel it's an important part of the story of polyA in prokaryotes. That being said, I would like for you to respond to the reviewers comments and update the manuscript accordingly. Please send me the revised manuscript once it's completed. I do not plan to send it back out for peer review, but will review it myself prior to making a final decision.

We thank the editor for endorsing our work and for the time and effort taken to help us improve it. Our answers to the reviewer's comments are below:

Reviewers' comments:

Reviewer #1 (Public repository details (Required)):

The authors performed RNA sequencing and have uploaded the results to a public database.

Reviewer #1 (Comments for the Author):

Summary

In this article, Xu et al. sequenced and analyzed the transcriptome of 2 pathogenic strains and 1 nonpathogenic strain of *Leptospira* through Oxford Nanopore Technologies' direct cDNA and direct RNA sequencing methods. They identified new operons, RNA molecules, and evidence of potential posttranscriptional polyadenylation in *Leptospira* transcriptomes. The use of ONT's sequencing may improve the understanding of leptospiral RNA landscapes and polyadenylation. Here, I have the several questions and comments.

Comments

1. In some parts of the text, the writing of 'leptospira's' should be replaced with 'leptospiral', such as line 36, line 59, and line 60.

We thank the reviewer for calling our attention to this detail. We have replaced all occurrences of '*Leptospira's*' in the current manuscript with 'leptospiral'.

2. line 72, a spelling mistake of 'INDRODUCTION' should be corrected.

We corrected the misspelling.

3. line 126, just a word of 'bacteria' is not enough to summarize the follow up contents of the method.

We changed the subtitle of the section to "*Leptospira* culture and RNA extraction".

4. line 131, 'a 4-day old culture' should be changed with 'a 4-day culture'.

Addressed.

5. line 127-line137, some important details are missed. Passage times of pathogenic *Leptospira*? Because the virulence of the pathogenic *Leptospira* decreases rapidly during in vitro passage. The number of leptospire used for RNA extraction.

We agree with the reviewer that it is common that *Leptospira* loses virulence in culture. However, the purpose of this work was to explore the use of Oxford Nanopore Technologies' direct cDNA and direct RNA sequencing methods for evaluating leptospiral transcriptome *in vitro*. To do so, we used strains that are continuously maintained in culture in the laboratory. In pathogenesis and virulence studies, we generally passage the bacteria in model systems such as hamsters and then use a low passage strain. In future studies we will be comparing the high and low passage strains to evaluate the differences between them.

6. The **differential RNA expression** detected between pathogenic and nonpathogenic leptospire should be further validated by qPCR.

We would like to emphasize that, despite the fact that we observed differences in transcriptome profiles between *Leptospira* strains, the goal of this study was not to evaluate the differential gene expression patterns, virulence or pathogenesis of *Leptospira*. From our experience, Nanopore technology is not an ideal methodology to evaluate differential gene expression patterns.

7. In addition to posttranscriptional polyadenylation modification, is there any other posttranscriptional modifications could be detected by ONT's RNA-Seq technologies?

Although not included in this study, we have tried to call methylation from sequences generated by ONT's RNA-Seq technologies. However, we were not able to obtain any methylation signal from this dataset.

8. In the figure legend of Figure S4, 'IC: *L. interrogans*' should be corrected with 'LIC: *L. interrogans*'.

We corrected 'IC: *L. interrogans*' to 'LIC: *L. interrogans*' in the caption of Figure S4.

Reviewer #2 (Public repository details (Required)):

The raw data had been deposited in a NCBI-SRA.

Reviewer #2 (Comments for the Author):

1. The English grammar should be revised for publication.

We carefully went through the manuscript correcting all grammar typos/mistakes. We also had a copy editor from our department reviewing the manuscript for any mistakes and address them as needed.

2. This research provided detailed transcriptome profiles of the pathogen *Leptospira*. However, the author did not emphasize their new finding in the main text figures.

We moved the Supplementary figures S5 and S6 to the main manuscript as main figures as Fig 4 and Fig 5 to showcase the other identified features of transcriptome profiles from different strains of *Leptospira* such as the identification of operons and unannotated genomic regions.

3. The major defect of the research design is that the pathogen was in vitro cultured, not the separated culture from cell or animal models. So this transcriptome may not reveal the pathogenesis of the pathogen in vivo. The author should mention the limitations in Discussion with references.

We would like to emphasize that the focus of this study was not on *Leptospira* pathogenesis or virulence. Our study provides the baseline information using a newly available long-read sequencing technology to study the RNA landscape of *Leptospira* and potentially other bacterial pathogens. We added a section

addressing the limitation of only using bacteria cultured *in vitro* in this study at the end of the discussion section (Line 497-500).

September 11, 2023

Dr. Sreekumari Rajeev
UT AgResearch
Biomedical and Diagnostic Sciences
2407 River Dr
Knoxville 37996

Re: Spectrum02234-23R1 (Leptospira transcriptome sequencing using long-read technology reveals unannotated transcripts and potential polyadenylation of RNA molecules)

Dear Dr. Sreekumari Rajeev:

Thank you for your excellent revisions to the manuscript. I look forward to seeing this paper published in Microbiology Spectrum, and hope it gains the attention it deserves from the research community, especially those working in the area of translational control of gene expression.

Your manuscript has been accepted, and I am forwarding it to the ASM Journals Department for publication. You will be notified when your proofs are ready to be viewed.

Sincerely,

Jonathan Jacobs
Editor, Microbiology Spectrum
